

# Effects of progessive *vs.* constant protocol whole-body vibration on muscle activation, pain, disability and functional performance in non-specific chronic low back pain patients: a randomized clinical trial

Tasneem Zafar[1], Saima Zaki[1], Md Farhan Alam[1], Saurabh Sharma[1], Reem Abdullah Babkair[2] and Shibili Nuhmani[3]

[1] Centre for Physiotherapy and Rehabilitation Sciences, Jamia Millia Islamia University, New Delhi, India
[2] Physiotherapy Department, Alhada Armed Forces Hospital, Alhada, Saudi Arabia
[3] Department of Physical Therapy, College of Applied Medical Sciences, Imam Abdulrahman Bin Faisal University, Dammam, Saudi Arabia

Corresponding author
Saurabh Sharma, ssharma@jmi.ac.in

## ABSTRACT

**Background and Objective.** Non-specific chronic low back pain (NSCLBP) is a prevalent condition causing significant disability and functional impairment. Whole-body vibration exercise (WBVE) has emerged as a new treatment method, but additional research is necessary to determine the optimal parameters of WBVE that would be beneficial for patients experiencing chronic low back pain (CLBP). This study aims to investigate the effects of two type of WBVE (constant *vs* progressive) on pain, disability, functional performance, and muscle activity in patients with NSCLBP.

**Methods.** Thirty-two individuals diagnosed with chronic low back pain (CLBP) without any specific causes were enrolled and randomly assigned to one of two intervention groups: a constant/fixed protocol WBVE group or a progressive protocol WBVE group. Participants underwent WBVE sessions for around 30 min, thrice weekly over a period of 8 weeks. Primary outcomes assessed included pain intensity, functional disability, functional performance, and electromyographic activity of core musculature, measured at baseline and upon completion of the intervention period. Temporal changes of each outcome variable across different periods and between groups were measured with repeated measures $2 \times 2$ mixed ANOVA. Further, the paired $t$-test was performed to compare pre- and post-treatment values within each group.

**Results.** Significant improvements were observed in both the constant and progressive WBVE protocol groups. Pain intensity decreased by 64.2% ($p < 0.001$) in the constant group and by 61.1% ($p < 0.001$) in the progressive group. Functional disability decreased by 48.1% ($p < 0.001$) in the constant group and by 53.3% ($p < 0.001$) in the progressive group. Functional performance improved by 16.5% ($p < 0.001$) in the constant group and by 16.9% ($p < 0.001$) in the progressive group. Electromyography (EMG) demonstrated significant improvements across all measured variables except % maximum voluntary isometric contraction (%MVIC) of external obliques (EO) in both intervention groups with time ($p < 0.001$). There was no statistically significant difference in the magnitude of improvement between the constant and progressive

WBVE protocols ($p > 0.05$), indicating both modalities' effectiveness in ameliorating CLBP symptoms and associated functional impairments.

**Conclusion**. The study demonstrates that both progressive and constant WBVE protocols are equally effective in reducing pain and disability in NSCLBP patients. These findings support the inclusion of progressive WBVE in clinical practice, offering a flexible treatment option that can be tailored to individual patient needs, ensuring both tolerability and effectiveness. This contributes valuable evidence towards optimizing WBVE protocols for managing NSCLBP.

## INTRODUCTION

Chronic low back pain (CLBP) represents a significant health concern within developed countries, contributing to extensive disability and imposing substantial economic burdens on healthcare systems (*Becker et al., 2010*; *Maher, Underwood & Buchbinder, 2017*). Estimates show that 7.5% of the global population suffers from CLBP, making it one of the leading causes of disability worldwide (*Hoy et al., 2014*). In the US alone, CLBP is estimated to cost over $200 billion annually in healthcare and lost productivity (*Cieza et al., 2021*). CLBP is characterized by persistent pain, diminished lumbosacral flexibility, compromised flexion-relaxation response, stiffness in lower back and impaired balance, which can exacerbate with physical activity (*Hoy et al., 2010*). Its pathophysiology is multifactorial, involving the muscular, connective, and neural systems (*Rittweger et al., 2002*). Initial injuries can lead to increased muscle tension and reduced circulation, perpetuating pain and immobility (*Linsiński, 2000*; *Rittweger et al., 2002*). Furthermore, weakening of the flexor muscles and imbalances in muscle strength have been identified as significant contributors to the development of CLBP (*Steele, Bruce-Low & Smith, 2014*).

Assessing muscle activation, particularly through electromyography (EMG), is a critical measure for evaluating muscle performance in individuals with CLBP (*Chen et al., 2019*). Previous Studies have shown that CLBP is associated with altered activation patterns in key stabilizing muscles, such as the erector spinae (ES) and lumbar multifidus (MF), potentially as both a cause and a consequence in CLBP (*Kuriyama & Ito, 2005*; *Steele et al., 2020*). Additionally, CLBP has been linked to delays in the activation of stabilizing muscles, including the ES and transverse abdominis (TrA), particularly during anticipatory movements (*Ferreira, Ferreira & Hodges, 2004*; *Hodges et al., 2003*; *Sadeghi et al., 2016*).

In the pursuit of efficacious management strategies for CLBP, an integration of stabilization and strengthening exercises was earlier favored (*Tian & Zhao, 2018*). Recent evidence suggests Pilates, motor control, resistance training, and aerobic exercise are effective interventions for CLBP (*Owen et al., 2020*). While occupational vibration has been considered a contributing factor to back pain, seated work positions may pose a greater risk (*Palmer et al., 2008*). Whole-body vibration exercise (WBVE) has emerged as a promising treatment, using controlled mechanical vibrations to improve muscle

strength, balance, and mobility (*Lau et al., 2011*). Unlike occupational vibration, which may contribute to back pain, WBVE is specifically designed to deliver controlled vibrations that can aid in the rehabilitation process.

WBVE induces rapid muscle length changes, triggering the tonic vibration reflex to periodically contract and relax muscles (*Roelants et al., 2006*). Additionally, WBVE increases circulation and tissue perfusion, aiding tissue recovery and reducing pain and inflammation (*Lohman 3rd et al., 2007*). Low-intensity WBV is generally preferred for muscle activation, though the interaction between vibration frequency, amplitude, and body position is crucial for optimal results (*Cardinale & Wakeling, 2005*; *Lam et al., 2016*). A study by *Cigdem Karacay et al. (2022)* demonstrated that WBVE at a frequency of 25 Hz with low amplitude of two mm, administered three times per week for 8 weeks, significantly improved pain and disability in individuals with CLBP. In recent study have explored frequencies ranging from 5 to 30 Hz for six sessions within 2 weeks and found reduced pain in older adults with musculoskeletal conditions (*Rüger et al., 2023*). The frequencies utilized in studies typically range from 20–50 Hz, with session durations of 3–12 min, conducted 2–3 times weekly (*Cigdem Karacay et al., 2022*; *Rüger et al., 2023*).

Although there is substantial research on WBV, few studies have systematically explored variations in WBV parameters to determine optimal training conditions. Indirect evidence, like the work of *Ritzmann et al. (2010)*, suggests a link between neuromuscular activation and frequency during WBV training (*Ritzmann et al., 2010*). However, it remains unclear whether a constant or progressive frequency yields better results. Additionally, most WBV protocols use fixed exercise durations, overlooking individual pain thresholds and the potential benefits of gradual progression. Comparing these two protocols is essential to optimize WBVE parameters, such as frequency and amplitude, for better outcomes. Understanding whether progressive WBVE offers added benefits over constant WBVE could improve clinical practice, enabling more tailored treatments and improving patient outcomes. This study aims to compare the effects of two distinct WBV protocols on key clinical outcomes in individuals with CLBP. Specifically, this study aims to assess and contrast the efficacy of these protocols in reducing pain, decreasing disability, enhancing functional performance, and increasing muscle activation in core stabilizing muscles. We hypothesize that a progressive WBV protocol (PP-WBV) will be as effective as a constant/fixed WBV (CP-WBV/FP-WBV) protocol in enhancing the aforementioned outcomes.

## METHODS

### Research design

This study is structured as a two-arm, parallel-group, randomized clinical trial incorporating an active control group, duly registered in the Clinical Trials Registry-India (CTRI) under the registration number (CTRI/2023/12/060897). In strict adherence to the ethical principles outlined in the Declaration of Helsinki (1964), this investigation received ethical clearance from the Institutional Human Ethical Committee of Jamia Millia Islamia, approval number 27/9/462/JMI/IEC/2023. Prior to their participation,

all subjects provided written informed consent, which included permission for the use of photographic documentation within the scope of this study. To protect participant confidentiality, data were anonymized using unique identifiers, and personal information was kept separate from the research data. Only authorized personnel had access to the data, ensuring compliance with data protection standards.

## Sample size calculation

The sample size for this randomized clinical trial was determined utilizing G*Power software version 3.1.9.4, predicated on an effect size of −1.04 observed in post-test VAS scores for non-specific low back pain, as detailed in previous research (*Wang et al., 2019*). To achieve a statistical power of 0.80 at an alpha level of 0.05, a total of 32 participants is requisite, allocating 16 individuals to each of the two comparative groups. The statistical analysis of the sample size calculation is conducted using the $t$-test for the difference between two independent means (two groups).

## Randomization and blinding process

Participants were randomly allocated in 1:1 ratio by computer-generated sequence generation employing the RAND function in Microsoft Excel. Allocation was concealed by a sealed envelope. This is a single-blinded clinical trial in which only the participants were blinded to the intervention.

## Participants

Participants suffering from CLBP without any specific cause were recruited for the study through a multifaceted approach that involved enlisting individuals directly from the Physiotherapy clinic at the Centre for Physiotherapy and Rehabilitation Sciences, Jamia Millia Islamia. Additionally, to broaden the participant pool, flyers were strategically placed in various public locations throughout New Delhi. Outreach efforts were further augmented by disseminating invitations through email and verbal communications, aiming to enhance participation in the research. These concerted efforts culminated in the successful recruitment of a total of 32 participants for the study. Patient flow is highlighted in the CONSORT flow diagram (Fig. 1).

Participants' Non-specific chronic low back pain (NSCLBP) was confirmed through a clinical examination conducted by a qualified physiotherapist, defined as a licensed professional with at least a bachelor's degree in physiotherapy, registered with the relevant governing body, and possessing a minimum of two years of clinical experience in musculoskeletal physiotherapy, particularly in the management of chronic low back pain, in addition to self-reported symptoms. This assessment included a detailed review of the patient's medical history, pain characteristics, and functional limitations. The examination ensured that the diagnosis met the established criteria for NSCLBP, defined as pain persisting for at least 12 weeks without any identifiable specific underlying cause.

The inclusion criteria for the study were participants aged from 30–60 years old, CLBP without any specific cause from 12 weeks at least and intermittent pain of three times a week over 3 months (*Balagué et al., 2012*). Body mass index (BMI) <24.5, and the patient should not have knee pain. Participants were excluded if they had undergone previous
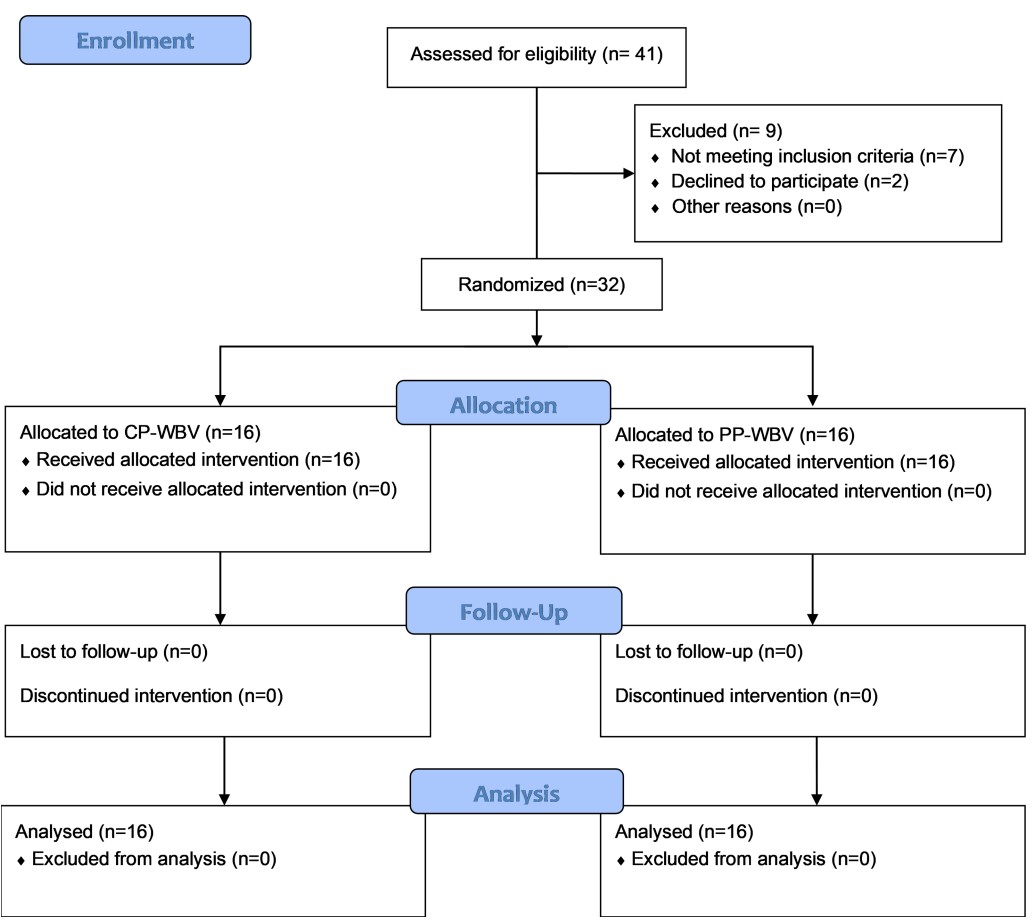

**Figure 1** Participants flowchart according to Consolidated Standards of Reporting Trials (CONSORT).

surgery, dislocation and fracture within the last 2 years, rheumatoid arthritis, ankylosing spondylitis and disc pathology, cardiovascular disease, progressive neuromuscular deficit, or severe osteoporosis (T-score $\leq -2.5$), pregnant or lactating, numerical pain scale $\geq 8$, attended WBV exercise in last 3 months (*Dong et al., 2020*). Participants were asked not to perform additional physical therapy interventions during the study period.

## Interventions

In this study, participants, unaware of their group assignment, were randomly allocated to either the CP-WBV or PP-WBV/FP-WBV groups using a computer-generated random number sequence. The WBV interventions were administered by experienced physical therapists on side-alternating type WBV platform by Crazy Fit VIVA Fitness following a standardized intervention protocol to minimize variability. Participants engaged in the exercise regimen for around 30 min per session, three times a week, over an eight-week period with amplitude of two mm in both the groups and fixed frequency of 18 Hz in constant protocol group and 5 Hz–20 Hz in progressive protocol group. Each session was structured to include a 5-min warm-up phase, consisting of active stretching exercises

**Table 1  Group 1 (Constant/Fixed protocol WBV).**

| Exercise program | Each Time | Repetition | Frequency | Rest | Total time |
|---|---|---|---|---|---|
| Squat | 90 s | 2 | 18 | 30 s | 180 s |
| Kneeling | 60 s | 2 | 18 | 30 s | 120 s |
| Bridge | 90 s | 2 | 18 | 30 s | 180 s |
| Bridge with leg lift | 60 s | 2 | 18 | 30 s | 120 s |
| Bridge with knee flex | 60 s | 2 | 18 | 30 s | 120 s |
| Back release | 90 s | 2 | 18 | 30 s | 180 s |

targeting the major muscles of the lower limb (slow sustained stretching for 3–5 repetitions per session, with each stretch held for 20–30 s and a 30-s rest interval between sets), followed by a series of six exercises performed on the WBV platform (*Wang et al., 2019*) (Fig. 2). The session concluded with a 5-min cool-down phase involving stretching exercises. Squat was performed at knee flexion of 30–45 degrees, kneeling with both hip and knee flexed to 90 degrees, bridge and knee flex by flexing knee at 90 degrees and back release by flexing trunk at 45 degrees with neutral spine alignment in each exercises. To maximize the transmission of vibrations and prevent damping, subjects were instructed to remove shoes on the WBV platforms during the exercises. Detailed protocols are delineated in the tables and figures, offering an exhaustive elucidation of the methodologies implemented in this study (Tables 1 and 2).

## Outcomes

Outcome measures in this study were administered by a single physiotherapist to maintain reliability and accuracy, with assessments conducted at baseline and post an eight-week intervention period by the same evaluator to ensure blinding integrity and measurement consistency. This study did not include follow-up assessments for the outcome variables beyond the post-intervention phase. The height (m) of the subject was measured using a MCP 2m/200CM Roll Ruler Wall Mounted Growth Stature Meter (Model No. 265M), with the subject standing against the wall and feet closed together without shoes. The weight (kg) of the subject was measured using the Omron Hn-289 Weighing Scale. The evaluation focused on four primary dependent variables: pain intensity, functional disability, functional performance, and core muscle activity. These were measured using a combination of validated instruments and methodologies, including the Visual Analog Scale (VAS) for pain intensity, the Roland Morris Disability Questionnaire (RMDQ) for functional disability, the Progressive Isoinertial Lifting Evaluation (PILE) test for functional performance, and the percentage of Maximum Voluntary Isometric Contraction (%MVIC) for assessing the activation of core muscles including the rectus abdominus (RA), external oblique (EO), ES, and MF.

### Visual analog scale

The VAS, employed for assessing pain intensity, is a subjective measure yet is recognized for its reliability and validity in capturing the pain experience of individuals. It features a 10 cm

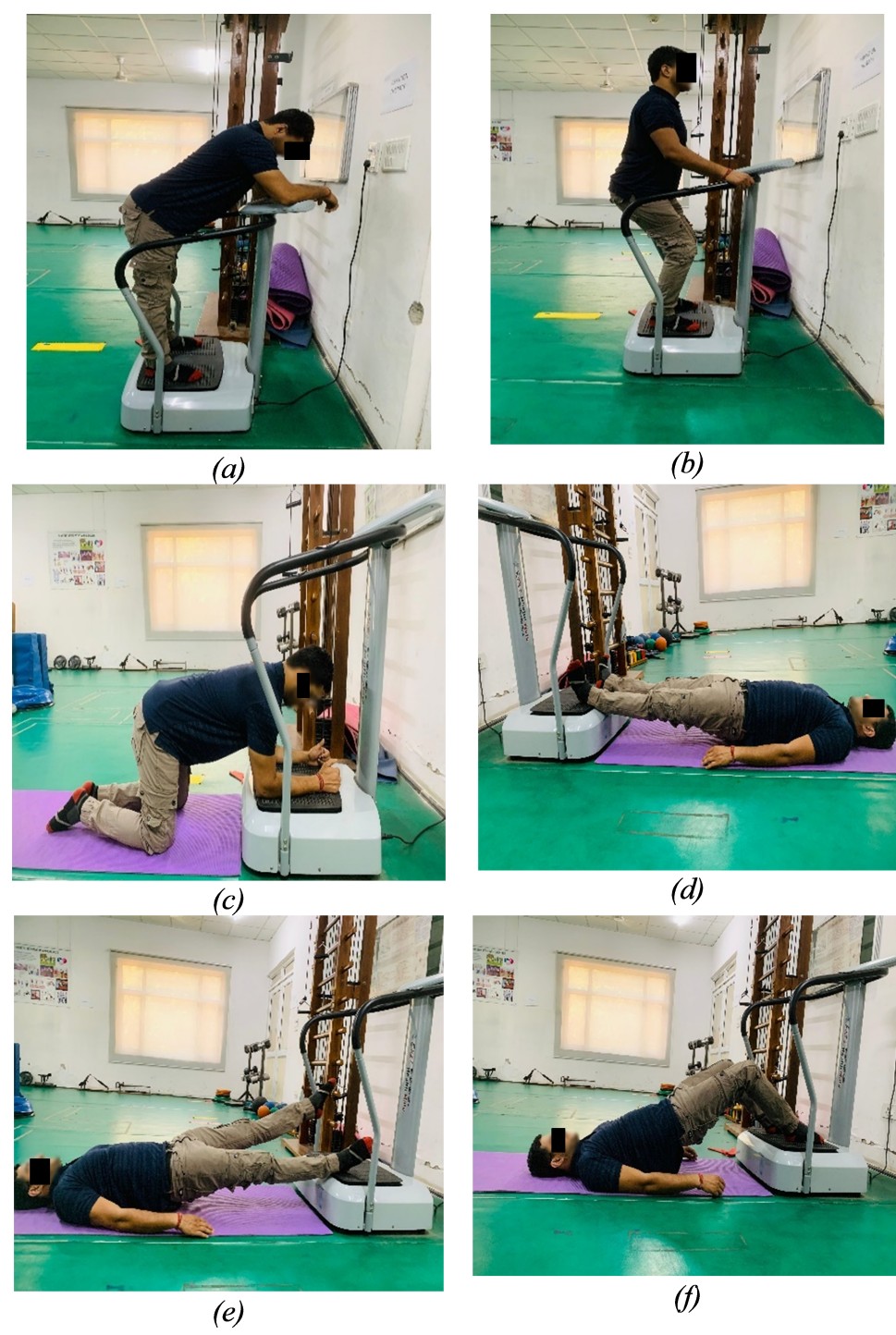

**Figure 2** Exercises: (A) Back release. (B) Squat. (C) Kneeling. (D) Bridge. (E) Bridge with leg lift. (F) Bridge with knee flex.

line representing a continuum from "no pain" (score of zero) to "extremely unbearable pain" (score of ten) (*Wewers & Lowe, 1990*). The minimal clinically important change
**Table 2  Group 2 (progressive protocol WBV).**

| Exercise program | Each time | Repetition | Frequency | Rest | Total time |
|---|---|---|---|---|---|
| Squat | First 2 weeks-30 s<br>Till 4 weeks-60 s<br>Till 8 weeks-90 s | 2 | First 2 weeks-5–12 Hz<br>Till 4 weeks-12–20 Hz<br>Till 8 weeks-20 Hz | 30 s | First 2 weeks-60 s<br>Till 4th week-120 s<br>Till 8th week-180 s |
| Kneeling | First 2 weeks-20 s<br>Till 4 weeks-40 s<br>Till 8 weeks-60 s | 2 | First 2 weeks-5–12 Hz<br>Till 4 weeks-12–20 Hz<br>Till 8 weeks-20 Hz | 30 s | First 2 weeks-40 s<br>Till 4th week-80 s<br>Till 8th week-120 s |
| Bridge | First 2 weeks-30 s<br>Till 4 weeks-60 s<br>Till 8 weeks-90 s | 2 | First 2 weeks-5–12 Hz<br>Till 4 weeks-12–20 Hz<br>Till 8 weeks-20 Hz | 30 s | First 2 weeks-60 s<br>Till 4th week-120 s<br>Till 8th week-180 s |
| Bridge with leg lift | First 2 weeks-20 s<br>Till 4 weeks-40 s<br>Till 8 weeks-60 s | 2 | First 2 weeks-5–12 Hz<br>Till 4 weeks-12–20 Hz<br>Till 8 weeks-20 Hz | 30 s | First 2 weeks-40 s<br>Till 4th week-80 s<br>Till 8th week-120 s |
| Bridge with knee flex | First 2 weeks-20 s<br>Till 4 weeks-40 s<br>Till 8 weeks-60 s | 2 | First 2 weeks-5–12 Hz<br>Till 4 weeks-12–20 Hz<br>Till 8 weeks-20 Hz | 30 s | First 2 weeks-40 s<br>Till 4th week-80 s<br>Till 8th week-120 s |
| Back release | First 2 weeks-30 s<br>Till 4 weeks-60 s<br>Till 8 weeks-90 s | 2 | First 2 weeks-5–12 Hz<br>Till 4 weeks-12–20 Hz<br>Till 8 weeks-20 Hz | 30 s | First 2 weeks-60 s<br>Till 4th week-120 s<br>Till 8th week-180 s |

(MCIC) for pain intensity on the VAS for patients with subacute or CLBP is identified as at least a 20 mm reduction (*Pires, Cruz & Caeiro, 2015*). Participants were instructed to indicate their perceived pain intensity on the scale using a pencil, with the scoring range established between 0 and 10, allowing for precise self-reported assessments of pain levels (*Wewers & Lowe, 1990*).

The VAS is widely acknowledged for its strong psychometric properties (*Alghadir et al., 2018*). Its validity in assessing pain intensity has been well-documented, demonstrating significant correlations with other measures of pain, such as the Numerical Rating Scale (NRS) (*Bijur, Latimer & Gallagher, 2003*). Furthermore, the VAS has demonstrated excellent test–retest reliability, particularly in patients with LBP, with Cohen's kappa ranging from 0.66 to 0.93 (*Roach et al., 1997*). This reliability underscores the consistency

of the VAS in measuring pain over repeated assessments, making it a reliable tool for capturing changes in pain intensity in clinical and research settings.

### Roland Morris disability questionnaire

The assessment of functional disability was conducted using the RMDQ, a modification of the Sickness Impact Profile specifically designed for functional disability measurement. The RMDQ scale ranges from 0, indicating no disability, to 24, with higher scores denoting severe disability. Participants are instructed to indicate "yes" or "no" for each item, depending on whether the statement applies to them on the day of assessment. As outlined by *Maughan & Lewis (2010)*, a mean difference of 5 on the RMDQ, constitutes the minimal clinically significant difference in patients with CLBP.

The RMDQ is a widely recognized tool for assessing functional disability in patients with LBP, known for its strong reliability and validity across various populations. The reliability of the RMDQ has been demonstrated in multiple studies. For example, the Turkish version of the RMDQ reported high internal consistency with Cronbach's alpha vsaalues exceeding 0.85, and an intraclass correlation coefficient (ICC) indicating high test–retest reliability (*Küçükdeveci et al., 2001*). Similarly, the Korean version of the RMDQ showed excellent internal consistency (Cronbach's alpha = 0.893) and an ICC of 0.837, further supporting its (*Kim & Lim, 2011*). Additionally, the Romanian version demonstrated an outstanding ICC of 0.95, reinforcing its test–retest reliability (*Ilie & Rusu, 2017*). In terms of validity, the RMDQ is well-validated for use in CLBP populations, with studies supporting its ability to accurately reflect changes in disability over time. These findings confirm that the RMDQ is a reliable and valid instrument for measuring disability in patients with CLBP.

### Progressive isoinertial lifting evaluation

For the evaluation of functional work performance and weight lifting capacity, the progressive isoinertial lifting evaluation (PILE) test was employed. This test provides a quantitative measure of functional strength and endurance, offering valuable insights into the effectiveness of the WBVE protocols in enhancing these physical capacities. This test required female participants to commence lifting at 5 lbs (approximately 2.5 kg) and male participants at 10 lbs (approximately 5 kg). The procedure involved lifting a box with weights from the floor to waist level and subsequently from waist to shoulder level, four times over a 30-second period. The test concluded upon reaching the participant's psychophysical limit (manifested as fatigue or fear), aerobic limit (indicated by achieving 85% of their maximum heart rate), or safety limit (exceeding 50% of their body weight) (*Mayer et al., 1988*).

The PILE test has demonstrated high test–retest reliability, with an intra-class correlation coefficient (ICC) of 0.91, indicating excellent consistency in assessing functional lifting capacity, particularly in patients with chronic lumbar pain (*Lygren et al., 2005*). The validity of the PILE test is well-established for measuring functional strength and endurance, making it a reliable tool for assessing the effectiveness of interventions aimed at improving physical capacity (*Mayer et al., 1988*; *Mohapatra, Verma & Girish, 2022*).

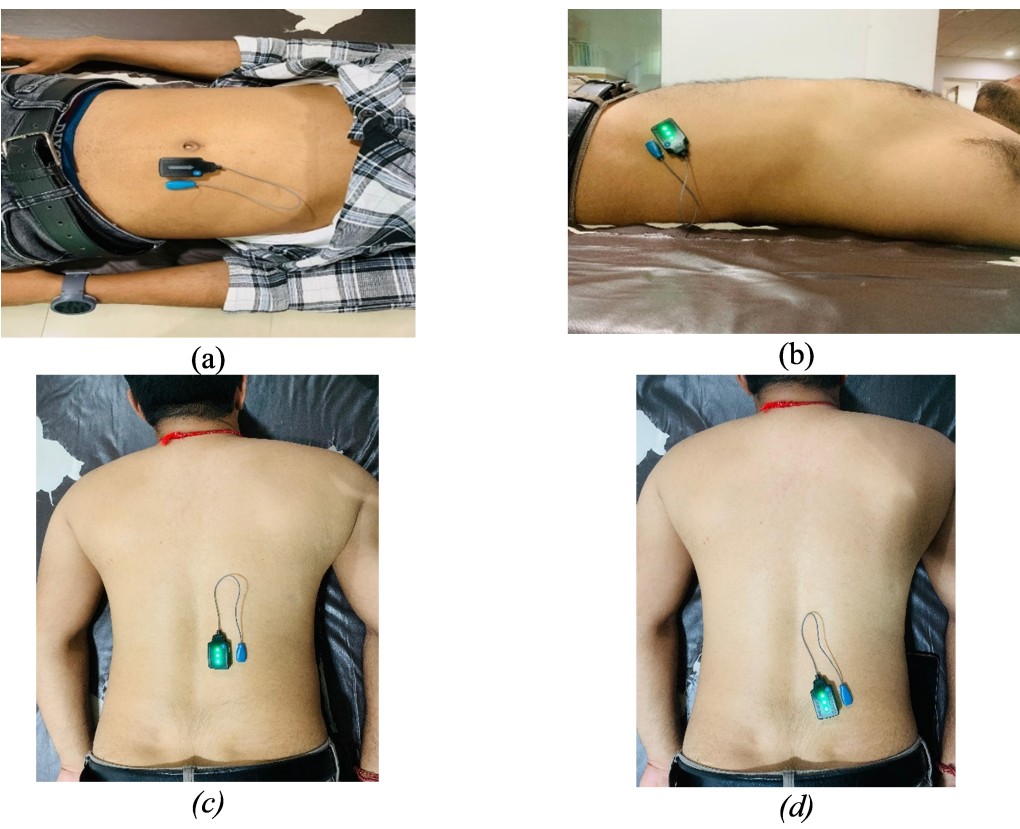

**Figure 3** Electrode placement: (A) rectus abdominis (B) external obliques (C) erector spinae (D) multifidus.

### Electromyography

EMG activity of the RA, EO, ES, and MF was captured utilizing the Delsys Trigno wireless EMG system, in conjunction with Lab Chart software version 7 from AD Instruments, New Zealand (Fig. 3). Surface EMG sensors facilitated the wireless connection of participants to the EMG system. Prior to sensor placement, the skin at each sensor location was meticulously prepared and cleansed to minimize electrical impedance, ensuring the accuracy and reliability of the EMG readings. The electrodes were positioned according to the guidelines outlined in the revised surface electromyography (sEMG) protocol (*Criswell, 2010*) (Fig. 3). Selection of lumbar and abdominal muscles for electrode placement was based on the side exhibiting more severe pain (*Ye, Ng & Yuen, 2014*).

For the MF, electrodes were positioned 2–3 cm from midline at the L5 level. In the case of the ES, placement was two cm away from the midpoint of a line connecting the bilateral iliac crests. For RA, the electrodes were located 2–3 cm lateral to the midline on the muscle's second segment. Finally, the EO electrodes were positioned at the midpoint of a line running from the anterior superior iliac spine to the tip of the 11th rib. This careful placement of electrodes was aimed at enabling precise assessments of muscle activity in

areas associated with significant pain, contributing to a comprehensive understanding of muscular dynamics under different conditions (*Ye, Ng & Yuen, 2014*).

The maximum voluntary isometric contraction (MVIC) measurements for all four muscles were taken to normalize the EMG amplitude and calculate % MVIC. After the MVIC testing, subjects were given a 5-min rest period. Subsequently, subjects were instructed to perform the same movement that was used during MVIC testing but without resistance to calculate the root mean square (RMS) in activity. They completed three trials of the movement, taking a 10-s rest between each trial.

$$\%MVIC = (RMS \text{ in normal activity}/RMS \text{ in MVIC}) \times 100.$$

## Statistical analysis

Statistical analyses were conducted utilizing IBM SPSS software, version 29.0.2.0 (IBM SPSS, Armonk, NY, USA). To evaluate the distribution normality of the variables, the Shapiro–Wilk test was employed. A $p$-value threshold of less than 0.05 was established to denote statistical significance. Independent $t$-tests or chi-squared tests were applied for the examination of demographic and baseline characteristics. Furthermore, to explore and compare the temporal changes of each outcome variable across different periods and between groups, a repeated measures 2X2 mixed ANOVA was implemented, incorporating factors of time, group, and the interaction between time and group. Further, the paired $t$-test was performed to assess significant changes by comparing pre- and post-treatment values within each group. The results, including $p$-values, were then visualized using graphs generated with GraphPad Prism version 10 (Fig. 4).

## RESULTS

A total of 41 participants were assessed for eligibility, out of which nine participants were excluded due to not meeting the eligibility criteria. Finally, total of 32 participants completed all of the exercise and assessment sessions having 16 participants in each group. The mean ages were $41.44 \pm 7.68$ years in the CP-WBV group and $39.94 \pm 6.78$ years in the PP-WBV group. No significant differences were detected in the baseline characteristics of both groups (Table 3).

### Pain

VAS for pain showed significant reductions over time ($F = 431, p < .001, \eta p^2 = 0.93$), with no significant differences between groups ($F = 0.00, p = 1.00, \eta p^2 = 0.00$) or time-group interactions ($F = 0.63, p = 0.43, \eta p^2 = 0.02$) (Fig. 4, Table 4).

### Disability

Disability, as measured by the RMDQ, improved significantly over time ($F = 888, p < .001, \eta p^2 = 0.96$). The interaction effect was significant ($F = 8.4, p = 0.007, \eta p^2 = 0.21$), although group effects were minimal ($F = 0.36, p = 0.55, \eta p^2 = 0.01$) (Fig. 4, Table 4).

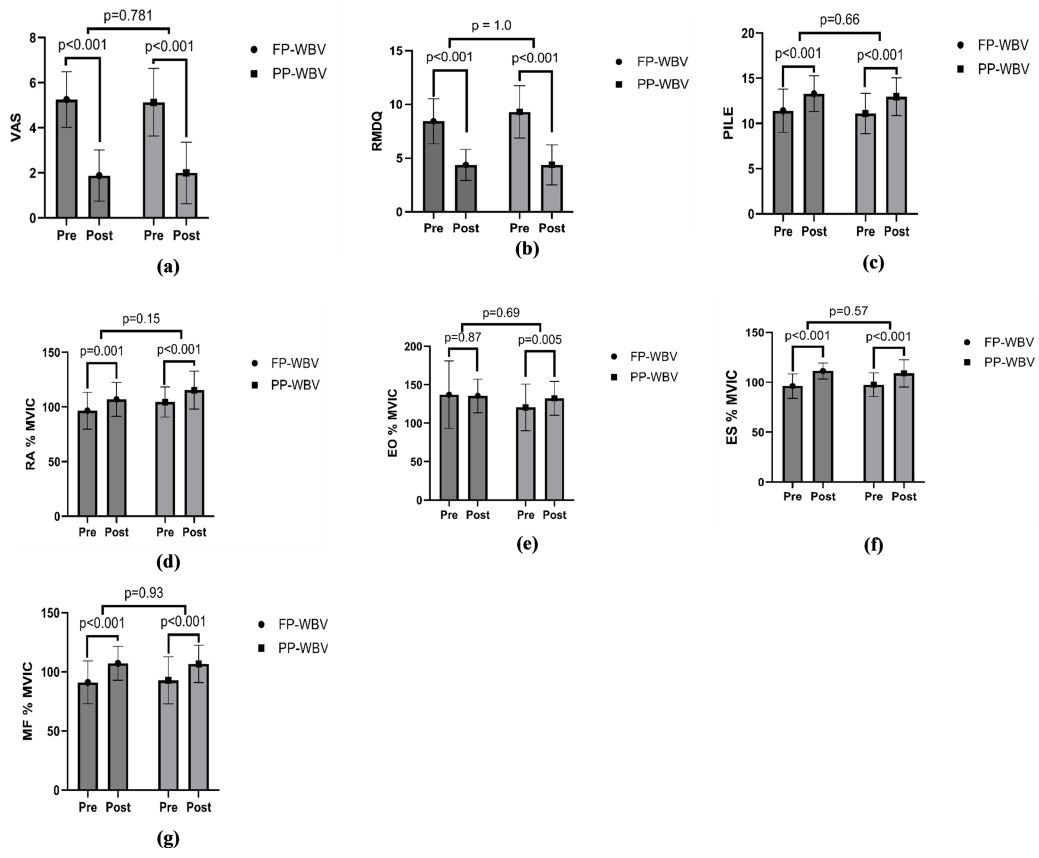

**Figure 4 Within group pre-post mean difference (A) VAS (B) RMDQ (C) PILE (D) RA %MVIC (E) EO %MVIC (F) ES %MVIC (G) MF %MVIC.** Abbreviations: VAS, Visual analog scale; RMDQ, Roland Moris disability questionnaire; PILE, Progressive isoinertial lifting evaluation; RA, Rectus abdominis; %MVIC, percentage maximal voluntary isometric contraction; EO, External oblique; ES, Erector spinae; MF, Multifidus.

## Functional performance

For functional performance, assessed through the PILE, showed significant improvement in performance over time ($F = 90$, $p < .001$, $\eta p^2 = 0.75$), with no significant group differences ($F = 0.17$, $p = 0.67$, $\eta p^2 = 0.006$) or interaction effects ($F = 0.00$, $p = 1.00$, $\eta p^2 = 0.00$) (Fig. 4, Table 4).

## Electromyography
### *Rectus abdominis*

RA average RMS, RA average MVIC and RA % MVIC were significantly increased over time in both the groups. Following 8 weeks of intervention the ANOVA revealed significant effect of time ($F = 23$, $p < 0.001$, $\eta p^2 = 0.44$). While the main effect of group ($F = 0.06$, $p = 0.79$, $\eta p^2 = 0.002$.) was insignificant as well as main effect of Time × Group ($F = 0.89$, $p = 0.35$, $\eta p^2 = 0.02$) was also insignificant for RA average RMS. For RA average MVIC, ANOVA revealed significant effect of time ($F = 17$, $p < .001$, and a moderate effect size, $\eta p^2 = 0.36$) while insignificant effects of Time × Group ($F = 1.36$, $p = 0.25$, $\eta p^2 = 0.04$)

**Table 3 Baseline characteristics of the patients.**

|  | CP/FP-WBV | PP-WBV | p value |
|---|---|---|---|
| Subjects ± n) | 16 | 16 | |
| Male/Female ± n) | 8/8 | 7/9 | 0.72 |
| Age ± years) | 41.44 ± 7.68 | 38.84 ± 5.68 | 0.56 |
| BMI ± kg/cm$^2$) | 21.32 ± 2.25 | 21.04 ± 1.99 | 0.71 |
| VAS | 5.25 ± 1.23 | 5.13 ± 1.50 | 0.79 |
| RMDQ | 8.44 ± 2.09 | 9.31 ± 2.44 | 0.28 |
| PILE | 11.40 ± 2.40 | 11.09 ± 2.23 | 0.70 |
| RA average RMS | 0.045 ± .008 | 0.043 ± .009 | 0.57 |
| RA average MVIC | 0.048 ± .011 | 0.041 ± .009 | 0.07 |
| RA % MVIC | 96.65 ± 16.8 | 104.62 ± 13.87 | 0.15 |
| EO average RMS | 0.047 ± .01 | 0.046 ± .01 | 0.73 |
| EO average MVIC | 0.037 ± .01 | 0.040 ± .01 | 0.43 |
| EO % MVIC | 137.1 ± 44.06 | 120.5 ± 30.3 | 0.22 |
| ES average RMS | 0.047 ± .014 | 0.049 ± .012 | 0.68 |
| ES average MVIC | 0.050 ± .015 | 0.051 ± .015 | 0.75 |
| ES % MVIC | 96.14 ± 12.39 | 97. 58 ± 11.92 | 0.74 |
| MF average RMS | 0.041 ± 0.014 | 0.041 ± 0.014 | 0.96 |
| MF average MVIC | 0.046 ± 0.017 | 0.046 ± 0.018 | 0.96 |
| MF % MVIC | 91.11 ± 18.1 | 92.78 ± 18.8 | 0.80 |

**Notes.**

Abbreviations: BMI, body mass index; VAS, Visual analog scale; RMDQ, Roland-Morris disability questionnaire; PILE, Progressive iso-inertial lifting evaluation; RA, Rectus abdominis; EO, External oblique; ES, Erector spinae; MF, Multifidus; RMS, root mean square; MVIC, maximum voluntary isometric contraction; n, no of participants; WBV, Whole-body vibration; CP/FP, Constant protocol/Fixed protocol; PP, Progressive protocol.

and between groups ($F = 1.7$, $p = 0.20$, $\eta p^2 = 0.05$). For RA % MVIC, ANOVA revealed significant effect of time $F = 55$, $p < .001$, $\eta p^2 = 0.65$ while insignificant effects of Time × Group ($F = 0.03$, $p = 0.86$, $\eta p^2 = 0.001$) and between groups ($F = 2.2$, $p = 0.14$, $\eta p^2 = 0.07$) (Fig. 4, Table 4).

## External oblique

EO average RMS and EO average MVIC were significantly increased over time in both the groups while EO % MVIC showed insignificant change. Following 8 weeks of intervention the ANOVA revealed significant effect of time ($F = 168$, $p < .001$, and a large effect size, $\eta p^2 = 0.84$). While the main effect of group ($F = 0.50$, $p = 0.48$, $\eta p^2 = 0.017$) was insignificant as well as main effect of Time × Group ($F = 1.53$, $p = 0.22$, $\eta p^2 = 0.04$) was also insignificant for EO average RMS. For EO average MVIC, ANOVA revealed significant effect of time ($F = 32$, $p < .001$, $\eta p^2 = 0.52$) while insignificant effects of Time × Group ($F = 3.09$, $p = 0.08$, $\eta p^2 = 0.09$) and between groups ($F = 0.00$, $p = 0.94$, $\eta p^2 = 0.00$). For EO % MVIC, ANOVA revealed insignificant effect of time ($F = 1.00$, $p = 0.32$, $\eta p^2 = 0.03$.), Time × Group ($F = 1.69$, $p = 0.20$, $\eta p^2 = 0.05$) and between groups ($F = 1.03$, $p = 0.31$, $\eta p^2 = 0.03$) (Fig. 4, Table 4).

**Table 4  Repeated measure ANOVA.**

| Variables | CP-WBV (n = 16) | | PP-WBV (n = 16) | | Source | p-value | ηp² | F |
|---|---|---|---|---|---|---|---|---|
| | Baseline mean ± SD | 8th week mean ± SD | Baseline mean ± SD | 8th week mean ± SD | | | | |
| VAS | 5.25 ± 1.23 | 1.88 ± 1.14 | 5.13 ± 1.50 | 2.00 ± 1.36 | T | <.001 | 0.93 | 431 |
| | | | | | T*G | 0.43 | 0.02 | 0.63 |
| | | | | | G | 1.00 | 0.00 | 0.00 |
| RMDQ | 8.44 ± 2.09 | 4.38 ± 1.45 | 9.31 ± 2.44 | 4.38 ± 1.86 | T | <.001 | 0.96 | 888 |
| | | | | | T*G | 0.007 | 0.21 | 8.4 |
| | | | | | G | 0.55 | 0.01 | 0.36 |
| PILE | 11.40 ± 2.40 | 13.28 ± 1.98 | 11.09 ± 2.23 | 12.96 ± 2.08 | T | <.001 | 0.75 | 90 |
| | | | | | T*G | 1.0 | 0.00 | 0.00 |
| | | | | | G | 0.67 | 0.006 | 0.17 |
| RA average RMS | 0.045 ± 0.008 | 0.058 ± 0.010 | 0.0438 ± 0.009 | 0.062 ± 0.024 | T | <.001 | 0.44 | 23 |
| | | | | | T*G | 0.35 | 0.02 | 0.89 |
| | | | | | G | 0.79 | 0.002 | 0.06 |
| RA average MVIC | 0.048 ± 0.011 | 0.054 ± 0.009 | 0.041 ± 0.009 | 0.052 ± 0.014 | T | <.001 | 0.36 | 17 |
| | | | | | T*G | 0.25 | 0.04 | 1.36 |
| | | | | | G | 0.20 | 0.05 | 1.7 |
| RA % MVIC | 96.65 ± 16.80 | 107.02 ± 15.43 | 104.62 ± 13.87 | 115.49 ± 17.35 | T | <.001 | 0.65 | 55 |
| | | | | | T*G | 0.86 | 0.001 | 0.03 |
| | | | | | G | 0.14 | 0.07 | 2.2 |
| EO average RMS | 0.047 ± 0.01 | 0.064 ± 0.01 | 0.046 ± 0.01 | 0.060 ± 0.01 | T | <.001 | 0.84 | 168 |
| | | | | | T*G | 0.22 | 0.04 | 1.53 |
| | | | | | G | 0.48 | 0.017 | 0.50 |
| EO average MVIC | 0.037 ± 0.01 | 0.048 ± 0.012 | 0.040 ± 0.01 | 0.046 ± 0.008 | T | <.001 | 0.52 | 32 |
| | | | | | T*G | 0.08 | 0.09 | 3.09 |
| | | | | | G | 0.94 | 0.00 | 0.00 |
| EO % MVIC | 137.1 ± 44.06 | 135.5 ± 21.96 | 120.58 ± 30.38 | 132.4 ± 21.98 | T | 0.32 | 0.03 | 1.00 |
| | | | | | T*G | 0.20 | 0.05 | 1.69 |
| | | | | | G | 0.31 | 0.03 | 1.03 |
| ES average RMS | 0.047 ± 0.01 | 0.061 ± 0.01 | 0.049 ± 0.01 | 0.060 ± 0.01 | T | <.001 | 0.81 | 134 |
| | | | | | T*G | 0.11 | 0.08 | 2.71 |
| | | | | | G | 0.96 | 0.00 | 0.002 |
| ES average MVIC | 0.050 ± 0.01 | 0.055 ± 0.014 | 0.51 ± 0.01 | 0.055 ± 0.009 | T | <.001 | 0.42 | 21.7 |
| | | | | | T*G | 0.34 | 0.30 | 0.91 |
| | | | | | G | 0.87 | 0.001 | 0.02 |

**Table 4** (*continued*)

| Variables | CP-WBV ($n = 16$) | | PP-WBV ($n = 16$) | | Source | *p*-value | $\eta p^2$ | F |
|---|---|---|---|---|---|---|---|---|
| | Baseline mean ± SD | 8th week mean ± SD | Baseline mean ± SD | 8th week mean ± SD | | | | |
| ES % MVIC | 96.14 ± 12.3 | 111.25 ± 8.08 | 97.58 ± 11.92 | 108.9 ± 13.7 | T | <.001 | 0.69 | 66.9 |
| | | | | | T*G | 0.26 | 0.04 | 1.31 |
| | | | | | G | 0.91 | 0.00 | 0.012 |
| MF average RMS | 0.041 ± 0.014 | 0.054 ± 0.014 | 0.041 ± 0.014 | 0.053 ± 0.011 | T | <.001 | 0.87 | 201 |
| | | | | | T*G | 0.40 | 0.02 | 0.72 |
| | | | | | G | 0.92 | 0.00 | 0.01 |
| MF average MVIC | 0.046 ± 0.017 | 0.051 ± 0.016 | 0.046 ± 0.018 | 0.051 ± 0.016 | T | <.001 | 0.66 | 60.2 |
| | | | | | T*G | 0.92 | 0.00 | 0.10 |
| | | | | | G | 0.97 | 0.00 | 0.01 |
| MF % MVIC | 91.11 ± 18.1 | 107.2 ± 14.3 | 92.78 ± 18.8 | 105.6 ± 15.8 | T | <.001 | 0.71 | 75.8 |
| | | | | | T*G | 0.54 | 0.01 | 0.37 |
| | | | | | G | 0.91 | 0.00 | 0.011 |

**Notes.**

Abbreviation: CP-WBV, Constant protocol whole-body vibration; PP-WBV, Progressive protocol whole-body vibration; VAS, Visual analog scale; RMDQ, Rolland Moris disability questionnaire; PILE, Progressive isoinertial lifting evaluation; RA, Rectus abdominis; EO, External oblique; ES, Erector spinae; MF, Multifidus; RMS, Root mean square; MVIC, maximum voluntary isometric contraction; SD, standard deviation; N, no of participants; T, time; G, group; T*G, Time group interaction.

## Erector spinae

Erector spinae (ES) average RMS, ES average MVIC and ES % MVIC were significantly increased over time in both the groups. Following 8 weeks of intervention the ANOVA revealed significant effect of time ($F = 134$, $p < .001$, and a large effect size, $\eta p^2 = 0.81$). While the main effect of group ($F = 0.002$, $p = 0.96$, $\eta p^2 = 0.00$) was insignificant as well as main effect of Time × Group ($F = 2.71$, $p = 0.11$, $\eta p^2 = 0.08$) was also insignificant for ES average RMS. For ES average MVIC, ANOVA revealed significant effect of time ($F = 21.7$, $p < .001$, $\eta p^2 = 0.42$) while insignificant effects of Time × Group ($F = 0.91$, $p = 0.34$, $\eta p^2 = 0.03$) and between groups ($F = 0.02$, $p = 0.87$, $\eta p^2 = 0.001$). For ES % MVIC, ANOVA revealed significant effect of time ($F = 66.9$, $p < .001$, and a large effect size, $\eta p^2 = 0.69$), while insignificant effects of Time × Group ($F = 1.31$, $p = 0.26$, $\eta p^2 = 0.04$) and between groups ($F = 0.012$, $p = 0.91$, $\eta p^2 = 0.000$) (Fig. 4, Table 4).

## Multifidus

Multifidus (MF) average RMS, MF average MVIC and MF % MVIC were significantly increased over time in both the groups. Following 8 weeks of intervention the ANOVA revealed significant effect of time ($F = 201$, $p < .001$, and a large effect size, $\eta p^2 = 0.87$). While the main effect of group ($F = 0.01$, $p = 0.92$, $\eta p^2 = 0.00$) was insignificant as well as main effect of Time × Group ($F = 0.72$, $p = 0.40$, $\eta p^2 = 0.02$.) was also insignificant for MF average RMS. For MF average MVIC, ANOVA revealed significant effect of time ($F = 60.2$, $p < .001$, and a large effect size, $\eta p^2 = 0.66$) while insignificant effects of Time × Group ($F = 0.10$, $p = 0.92$, $\eta p^2 = 0.00$) and between groups ($F = 0.01$, $p = 0.97$, $\eta p^2 = 0.00$). For MF % MVIC, ANOVA revealed significant effect of time $F = 75.8$, $p < .001$, and a large

effect size, $\eta p^2 = 0.71$, while insignificant effects of Time $\times$ Group ($F = 0.37$, $p = 0.54$, $\eta p^2 = 0.01$) and between groups ($F = 0.01$, $p = 0.91$, $\eta p^2 = 0.00$) (Fig. 4, Table 4).

## DISCUSSION

The present study aimed to comprehensively evaluate the effects of 8-week CP-WBV *vs* 8-week PP-WBV on patients with CLBP. The results showed that both these interventions improved the pain intensity and disability of CLBP after the intervention. Further, CP-WBV and PP-WBV enhanced the contractility of deep trunk muscles and improve functional performance in patients with CLBP.

In this investigation, CP-WBV yielded results analogous to PP-WBV regarding the reduction of pain and the improvement of disability. This equivalence in outcomes can be attributed to the strategic progression of the frequency and duration in PP-WBV treatments to align with those of CP-WBV, especially targeting the optimal effects observed at a frequency of 20 HZ, as suggested by existing research (*Del Pozo-Cruz et al., 2011*). This approach underscores a potential clinical application where practitioners could incrementally adjust the intensity of treatment in harmony with the patient's tolerance levels. Such a methodology is supported by previous findings, which have indicated that a progressively implemented WBV protocol can lead to significant advancements in managing CLBP (*Sajadi et al., 2019*).

First, the diminution of pain through vibration could be attributed to the inhibition of smaller fiber-mediated pain signals (specifically A-$\delta$ or C fibers) during their transmission to the central nervous system, coupled with the stimulation of larger A-$\beta$ fibers, thereby effectuating the closure of the pain gate mechanism and effectively reducing the perception of pain (*Wang et al., 2019*). Secondly, the enhancement of pain associated with muscle tension might be facilitated by the induction of muscle relaxation *via* vibratory stimuli (*Elfering et al., 2016*). Lastly, the enhancement of posture through the activation of trunk muscles may contribute to the alleviation of undue tension and mechanical stress on the trunk's passive structures.

The reduction of disability facilitated by WBVE is attributed to several physiological pathways. Primarily, WBVE enhances muscle function through increased activation of muscle fibers and improved recruitment patterns. This augmentation is critical for stabilizing the lumbar spine and reducing back mechanical stress (*Rittweger, Mutschelknauss & Felsenberg, 2003*). Strengthening muscle power and enhancing coordination enhance functional performance, which may reduce disability associated with CLBP. Furthermore, WBVE has been shown to improve proprioceptive feedback, essential for maintaining balance and stability during movement (*Zafar et al., 2024*). Improved proprioception leads to better motor control, reducing the risk of falls and injuries, thereby contributing to lower disability scores (*Winter et al., 2022*).

WBVE has been demonstrated to significantly enhance muscle performance and strength (*Perchthaler, Grau & Hein, 2014*; *Torvinen et al., 2002*). This enhancement is primarily attributed to the activation of a wider array of the motor neuron pool, as well as the incorporation of previously inactive motor units into contraction, leading to a more

effective utilization of force production capabilities (*Baard, Pietersen & Rensburg, 2011*; *Torabi et al., 2013*). This outcome is closely linked to the induction of the tonic vibration reflex (TVR), which alters the length of muscle spindles, thereby potentially increasing muscle activity and improving the efficiency of the lumbar flexors and extensors (*Hagbarth, Hongell & Wallin, 1970*). Moreover, WBV exercise is believed to improve proprioception and augment muscle coordination in the lumbar region through the stimulation of mechanoreceptors in the lumbo-pelvic area (*Yang et al., 2015*). Studies have also shown a significant increase in the endurance of abdominal and multifidus muscles in patients with CLBP following WBV exercise. The heightened recruitment of motor units is thought to lead to more efficient force production, particularly during lifting activities. Previous research has indicated that WBV outperforms a control group in reducing pain intensity scores and functional performance, but it is not superior to the effectiveness of the core stabilization group (*Cigdem Karacay et al., 2022*).

### Strength and limitation

To the best of our knowledge, this is the first interventional study to compare the effects of PP-WBV in relation to CP-WBV, offering valuable insights for clinical practice in the treatment of CLBP. Moreover, this study introduces a progressive protocol specifically designed for core strengthening, diverging from previous progressive protocols that predominantly focused on balance and proprioception. Unlike prior research, where EMG activation was measured at a single point, our study provides insights into EMG activity following an 8-week training regimen. Lastly, the adherence to standardized protocols within this study was meticulous, ensuring methodological consistency and reliability throughout the research process. While our study provides valuable insights, it also has some limitations. First, we did not evaluate the follow-up effects of both WBV protocols on CLBP. The absence of a passive control group prevented us from comparing the individual effects of each protocol against a no-intervention scenario, limiting our ability to distinctly attribute observed improvements solely to the WBV protocols. Muscle activity was measured using surface electrodes, which could introduce variability due to factors such as electrode placement and skin impedance, potentially affecting the accuracy of the readings.

### Future prospective

Advancing CLBP research will benefit from follow up studies and larger sample size with passive control group, focusing on outcomes like strength, gait, sleep quality, and work productivity to improve treatment strategies and patient well-being. This protocol can be tested on different pattern of WBV like horizontal or triaxial WBV to see whether they produce equivalent results.

## CONCLUSION

In conclusion, the results of this study indicate that both CP-WBV and PP-WBV significantly reduce pain and disability while also improving the activity of the RA, EO, ES, and MF muscles, thus enhancing functional performance. These outcomes underscore that

PP-WBV is as effective as CP-WBV in the treatment of NSCLBP, presenting viable options for clinical intervention with the flexibility to tailor protocols to individual patient needs.

## ACKNOWLEDGEMENTS

The authors would like to extend their heartfelt gratitude to all the participants who enrolled in this study. Your dedication, time, and commitment have been invaluable to the success of our research. Without your enthusiastic involvement and cooperation, this study would not have been possible.

### Funding
The authors received no funding for this work.

### Competing Interests
The authors declare there are no competing interests.

### Author Contributions
- Tasneem Zafar conceived and designed the experiments, performed the experiments, analyzed the data, prepared figures and/or tables, authored or reviewed drafts of the article, and approved the final draft.
- Saima Zaki conceived and designed the experiments, performed the experiments, analyzed the data, prepared figures and/or tables, authored or reviewed drafts of the article, and approved the final draft.
- Md Farhan Alam conceived and designed the experiments, authored or reviewed drafts of the article, and approved the final draft.
- Saurabh Sharma conceived and designed the experiments, performed the experiments, prepared figures and/or tables, and approved the final draft.
- Reem Abdullah Babkair performed the experiments, authored or reviewed drafts of the article, and approved the final draft.
- Shibili Nuhmani analyzed the data, prepared figures and/or tables, and approved the final draft.

### Ethics
The following information was supplied relating to ethical approvals (i.e., approving body and any reference numbers):

Institutional Human Ethical Committee of Jamia Millia Islamia (27/9/462/JMI/IEC/2023).

### Data Availability
The raw measurements are available in the Supplementary Files.

**Clinical Trial Registration**

The following information was supplied regarding Clinical Trial registration:

Clinical Trials Registry-India (CTRI) under the registration number [CTRI/2023/12/060897]

**Supplemental Information**

Supplemental information for this article can be found online at http://dx.doi.org/10.7717/peerj.18390#supplemental-information.

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
