# Peer review of "Effects of progessive vs. constant protocol whole-body vibration on muscle activation, pain, disability and functional performance in non-specific chronic low back pain patients: a randomized clinical trial"

_PeerJ, doi:10.7717/peerj.18390_

## Round 0.1 · original submission · Major Revisions

Dear Dr. Sharma,

Thank you for submitting your manuscript to PeerJ. After careful consideration and review by our expert reviewers, we have identified several critical areas that need substantial revision before your manuscript can be considered for publication. Your manuscript was reviewed by three reviewers, and their feedback highlights several important concerns.

Please address all concerns raised by the reviewers and my comments specifically and in detail. Failure to fully address these concerns may result in the rejection of your manuscript. However, I believe you have the potential to revise your manuscript sufficiently and comprehensively.

We look forward to receiving your revised manuscript.

Summary of Required Revisions

Title
• The title should be revised for greater specificity to reflect the comparison between constant and progressive WBVE protocols.
General Comments
Terminology: It is recommended to use the term "Non-specific Low Back Pain (NSLBP)" for clarity and specificity.
Abstract: In the first line of the abstract, it is preferable to avoid statements that require citations. Instead, consider stating, "NSLBP is a prevalent condition causing significant disability.
Method
Recommended change: Thirty-two individuals with CLBP were randomly assigned to constant (20Hz) or progressive (5-20Hz) WBVE protocols. Participants underwent 30-minute WBVE sessions thrice weekly for 8 weeks. Outcomes included pain intensity (VAS), functional disability (RMDQ), functional performance (PILE test), and electromyographic activity of core muscles.
In the result section of the abstract you need to include the statistical test used and percentage of change.
Recommended change: Significant improvements were observed in pain intensity (VAS decreased by xx%, p < 0.001), functional disability (RMDQ decreased by xx%, p < 0.001), and muscle activity (MVIC increased by xx%, p < 0.001) in both groups. No significant differences were found between the constant and progressive protocols (p > 0.05).
Introduction
Line 46-48: Include recent data on the prevalence and economic burden of CLBP for context.
Line 49-51: Expand on the specific symptoms and impacts of CLBP.
Line 52-58: Simplify the language regarding the pathophysiology of CLBP for clarity.
Line 59-66: Summarize key findings on muscle activation and EMG studies more succinctly.
Line 67-72: Focus more on the relevance of WBVE in exercise interventions for CLBP.
Line 73-79: Add more details on the proposed mechanisms of WBVE.
Line 80-86: Provide specific examples of studies with varying WBVE parameters and their outcomes.
Line 87-92: Emphasize the rationale for comparing constant and progressive WBVE.
Line 93-96: State the study aim with more detail and clarity.
Methods
Line 97-100: Clearly define the sample size and provide justification for the chosen number of participants.
Line 101-105: Ensure detailed ethical considerations, including participant confidentiality and data protection.
Line 106-114: Provide more information on the recruitment process and inclusion/exclusion criteria.
Line 115-121: Justify the specific WBVE frequencies and durations used in the study.
Line 122-129: Describe the randomization process in more detail to ensure reproducibility.
Line 130-138: Standardize the intervention protocols and provide references to similar studies.
Line 139-150: Include detailed descriptions of the outcome measures and their validity.
Line 151-157: Explain the procedures for measuring pain intensity with the Visual Analog Scale (VAS).
Line 158-167: Clarify the criteria and significance of the Roland Morris Disability Questionnaire (RMDQ).
Line 168-175: Elaborate on the Progressive Isoinertial Lifting Evaluation (PILE) test and its relevance.
Line 176-198: Provide detailed procedures for electromyography (EMG) measurements and placement.
Line 199-212: Discuss the sample size calculation methodology in more detail.
Line 213-217: Explain the blinding process and how it was maintained throughout the study.
Line 218-227: Include details on statistical analysis methods and software used.
Results
Line 228-233: Clearly present the demographic characteristics and ensure tables are not overly dense.
Line 234-237: Provide specific numerical results and p-values for the pain reduction outcomes.
Line 238-241: Include detailed statistical data for the disability improvement outcomes.
Line 242-245: Present specific results for functional performance and their significance.
Line 246-257: Summarize the electromyography findings with clear statistical data.
Line 258-268: Discuss non-significant findings and provide interpretation.
Line 269-280: Provide specific data for the changes in erector spinae and multifidus muscle activity.
Discussion
Line 281-292: Interpret the results in the context of existing literature and discuss clinical implications.
Line 293-300: Compare findings with previous studies and explain any discrepancies.
Line 301-313: Discuss the mechanisms behind the observed effects of WBVE on CLBP.
Line 314-322: Elaborate on the physiological pathways through which WBVE improves disability.
Line 323-337: Provide a detailed discussion on the impact of WBVE on muscle performance and coordination.
Line 338-345: Acknowledge the limitations, including sample size and lack of a passive control group.
Line 346-354: Highlight the strengths of the study, including the novelty and rigorous methodology.
Line 355-359: Suggest specific directions for future research to address identified limitations.
Conclusion
Line 360-365: Summarize the key findings concisely and emphasize their clinical significance.

Reviewer 1 ·

Basic reporting

There are additional segments like strength of study, should be excluded or merged with other appropriate segment of the article

Experimental design

No Comment

Validity of the findings

No comment

Additional comments

Rationale should be convincing
Review the ITT analysis protocol
Re-check the appropriate statistical tests for study design
Re-write the limitations
Re-write the conclusion of the abstract in a clearly manner

Annotated reviews are not available for download in order to protect the identity of reviewers who chose to remain anonymous.

Reviewer 2 ·

Basic reporting

The written language of the research is fluent and clear. The research does not require any language services
 The reason for the research was supported by current literature information. Adequate background has been created about the patient group that constitutes the study population and WBV, which is the main treatment method.
 Standard Sections; The abbreviation MVIC should be stated at the first mention in the abstract section.
 The figures are relevant to the subject and their labels are stated appropriately.
 Raw data is presented clearly. The data is stated in the excel file to match the findings.

Experimental design

The study was planned to investigate the optimal treatment technique in CLBP patients and focused on the difference of exercise training in choosing the appropriate treatment technique.
 The treatment procedures used for the study do not pose ethical problems.
 The research methodology is presented in detail, subsequent researchers can use this methodology as a reference. However, the shortcomings of the research method are stated in the general comments section.

Validity of the findings

The research data is ready for use and the method of statistical analysis of the data has been validated.
 The conclusion section answers the research question to a high extent. However, in line 362, an increase in strength in certain muscles is mentioned; the change in the electrical activity of the muscle does not give an accurate idea about the strength of that muscle. If the strength increase here is interpreted according to PILE, this test method cannot be interpreted only for RA, EO, ES, and MF muscles. With this test, other global muscles are also included. Therefore, it should be avoided to make any comments about the strength increase of the core muscles.

Additional comments

How participants with CLBP received this diagnosis should be stated in the material and methods section. Was this decided as a result of the participant's self-report or any clinical examination?
 Give a detailed introduction of the WBV device used in the study. (side-alternating whole body vibration or vertical ? ) It has been reported that the effectiveness of devices can vary depending on the way they generate vibration. Please specify in the material method section
 What was the amplitude value during WBV training and was it the same in both groups? Please specify in the material method section.
 One of the most important criteria affecting the research result is the selected vibration frequency. 18 Hz vibration frequency was selected for the 1st group and a progressive vibration frequency was selected for the 2nd group. It should be stated on what basis these frequency ranges were chosen. Reference articles regarding this should be cited in the research. In most of the studies in the literature, frequencies above 20 Hz were selected. However, in this study, even the 5-12 Hz option was used. The different use of frequency in both groups may suggest that the results may be due to vibration frequency, not exercise. These parts need to be explained in detail based on the literature.
 Exercise definitions should be made in more detail so that the exercises can be applied in a standardized way. For example, at what degree of flexion angle were the knees fixed during the squat, or were the exercises performed dynamically or statically? Descriptive information for the exercises can be stated in the material method or exercise figures section.
 The validity and reliability of Progressive isoinertial lifting evaluation (PILE) should be stated in the material method section.
 Line 243 includes a statement about pain reduction in the functional performance findings section. I recommend checking that section again.
 In line 295, the word "strength" should be removed.
 In line 323, increased muscular performance, more motor unit synchronization and activation are appropriate expressions, but it is not possible to talk about an increase in strength. To show the increase in strength, it would be necessary to use isokinetic dynamometers with trunk attachment or core endurance tests. Surface EMG reflects the amount of musculoskeletal electrical activity in a noninvasive manner. Although this signal is related to muscle strength, it cannot be used directly as an expression of strength.

Annotated reviews are not available for download in order to protect the identity of reviewers who chose to remain anonymous.

·

Basic reporting

- Language and Professionalism: The manuscript is written in clear, professional, and technically correct English. The text is unambiguous and maintains a high standard of courtesy and expression throughout.
- Literature References and Background: The introduction provides a comprehensive background on chronic low back pain (CLBP) and the emerging use of whole-body vibration exercise (WBVE) as a treatment method. The manuscript appropriately references relevant prior literature, offering a well-rounded context for the study.
- Structure, Figures, Tables, and Raw Data: The manuscript follows a standard scientific structure, including sections such as Introduction, Methods, Results, and Discussion.
- Self-Contained and Relevant Results: The manuscript is self-contained and presents results that are directly relevant to the hypotheses stated. The study compares the effects of two WBVE protocols, providing a clear and thorough analysis of their impact on CLBP patients.

Experimental design

- Original Research and Scope: The study is original and falls within the scope of investigating treatment methods for CLBP. The research question is well-defined, focusing on the comparative effectiveness of constant vs progressive WBVE protocols.
- Rigorous Investigation: The investigation is conducted rigorously, with a clear description of the research design, including randomization and ethical approval. The study adheres to high technical and ethical standards.
- Methodology: The methods are described in detail, providing sufficient information for reproducibility. The intervention protocols, outcome measures, and statistical analyses are well-documented, ensuring that another investigator could replicate the study.

Validity of the findings

- Impact and Novelty: The study contributes valuable insights into the effectiveness of WBVE protocols for CLBP. While the impact and novelty are not explicitly assessed, the manuscript highlights the clinical relevance of the findings.
- Conclusions: The conclusions are well-stated, linked to the original research question, and supported by the results. The study concludes that both WBVE protocols are effective in managing CLBP, with no significant difference in efficacy between the constant and progressive protocols.

---

## Round 0.2 · Minor Revisions

Dear Authors,

I hope this message finds you well.

Thank you for the amendments you have made to the manuscript based on the reviewers' suggestions. We appreciate your attention to detail in addressing their feedback.

At this stage, I kindly ask you to respond to the remaining minor comments raised by Reviewer 1. Please ensure that all points are addressed thoroughly to facilitate the next steps in the review process.

We look forward to receiving your response.

Best regards,
Faizan Zaffar Kashoo

Reviewer 1 ·

Basic reporting

Reduce the length of Introduction.

Experimental design

The previous literature reported the large effect size so what the authors wanted to study? The authors should review the power (might be bigger than the 0.80) to justify the efficiency of the sample.
Line 163: What was the operational definition of Qualified physiotherapist? advise to add it.
Additionally, was there same physiotherapist for the assessments/screening of samples with LBP? advise to elaborate it too.
Line 169: Add the reference of CLBP criteria
Line 171: better to disclose the duration of such disorders....like surgery in last 12 months etc
Line 173: Does the author mean only sever osteoporosis exclude in the study and mild/moderate osteoporosis, normally called osteopenia, included??
Line 197: How many physiotherapists were involved? Ideally, the same, single physiotherapist assessed all the participants to avoid the biasedness.
Line 296: Advise to justify the use of ANOVA family? if there are two assessment times.
Line 298-99: Add the post hoc analysis for within the group analysis as well

Validity of the findings

Correlate with the Experimental design.

Additional comments

All above comments

Annotated reviews are not available for download in order to protect the identity of reviewers who chose to remain anonymous.

Reviewer 2 ·

Basic reporting

I have previously expressed my opinions regarding this section and I found the corrections made accordingly appropriate.

Experimental design

I have previously expressed my opinions regarding this section and I found the corrections made accordingly appropriate.

Validity of the findings

I have previously expressed my opinions regarding this section and I found the corrections made accordingly appropriate.

Additional comments

The revision suggested to the authors has been completed successfully.

·

Basic reporting

Basic reporting meets required criteria.

Experimental design

Design of the study meets the standards

Validity of the findings

Findings in the manuscript is valid.

---

## Round 0.3 · accepted · Accept

Thank you for submitting the revised manuscript. I have thoroughly reviewed the revised version and I'm pleased to confirm that all reviewers' comments have been adequately addressed. I find the manuscript to be in excellent shape and ready for publication.

Reviewer 1 ·

Basic reporting

Opinions advised has been addressed.

Experimental design

Opinion advised has been addressed.

Validity of the findings

Opinions has been incorporated.

Additional comments

Opinions has been addressed